# Exploring the Binding Interaction of Active Compound of Pineapple against Foodborne Bacteria and Novel Coronavirus (SARS-CoV-2) Based on Molecular Docking and Simulation Studies

**DOI:** 10.3390/nu14153045

**Published:** 2022-07-25

**Authors:** Mohammed F. Abuzinadah, Varish Ahmad, Salwa Al-Thawdi, Shadi Ahmed Zakai, Qazi Mohammad Sajid Jamal

**Affiliations:** 1Department of Medical Laboratory Technology, Faculty of Applied Medical Sciences, King Abdulaziz University, Jeddah 21589, Saudi Arabia; mabuzinadah@kau.edu.sa; 2Health Information Technology Department, The Applied College, King Abdulaziz University, Jeddah 21589, Saudi Arabia; 3Department of Biology, University of Bahrain, Sakhir Campus, Sakhir P.O. Box 32038, Bahrain; salthawadi@uob.edu.bh; 4Department of Medical Microbiology and Parasitology, Faculty of Medicine, King Abdulaziz University, Jeddah 21589, Saudi Arabia; szakai@kau.edu.sa; 5Department of Health Informatics, College of Public Health and Health Informatics, Qassim University, Al Bukayriyah 52741, Saudi Arabia; m.quazi@qu.edu.sa

**Keywords:** bromelain, antibacterial, membrane protein, docking studies, molecular dynamics simulation

## Abstract

Natural resources, particularly plants and microbes, are an excellent source of bioactive molecules. Bromelain, a complex enzyme mixture found in pineapples, has numerous pharmacological applications. In a search for therapeutic molecules, we conducted an in silico study on natural phyto-constituent bromelain, targeting pathogenic bacteria and viral proteases. Docking studies revealed that bromelain strongly bound to food-borne bacterial pathogens and SARS-CoV-2 virus targets, with a high binding energy of −9.37 kcal/mol. The binding interaction was mediated by the involvement of hydrogen bonds, and some hydrophobic interactions stabilized the complex and molecular dynamics. Simulation studies also indicated the stable binding between bromelain and SARS-CoV-2 protease as well as with bacterial targets which are essential for DNA and protein synthesis and are required to maintain the integrity of membranous proteins. From this in silico study, it is also concluded that bromelain could be an effective molecule to control foodborne pathogen toxicity and COVID-19. So, eating pineapple during an infection could help to interfere with the pathogen attaching and help prevent the virus from getting into the host cell. Further, research on the bromelain molecule could be helpful for the management of COVID-19 disease as well as other bacterial-mediated diseases. Thus, the antibacterial and anti-SARS-CoV-2 virus inhibitory potentials of bromelain could be helpful in the management of viral infections and subsequent bacterial infections in COVID-19 patients.

## 1. Introduction

Infectious diseases are one of the most dangerous burdens on human health because of the failure of first-line antibiotic-based chemotherapy. Moreover, an increase in resistance against prescribed antibiotics in many bacteria, including *Pseudomonas aeruginosa* (*P. aeruginosa)*, *Enterococcus faecium* (*E. faecium*), *Salmonella* spp., *Campylobacter*, *Streptococcus pneumonia* (*S. pneumonia*), *Neisseria gonorrhoeae* (*N. gonorrhoeae*), etc., has been reported. Bromelain recovered from pineapple fruit is a type of protein-degrading enzyme. The protein bromelain purified from the stem is made in a different way and has a different enzymatic makeup, such as peroxidase and cellulase, and enzyme inhibitors are included. The therapeutic potential of bromelain has been reported, with it having antiedematous, fibrinolytic, antithrombotic, antibacterial, and anti-inflammatory properties in both in vitro and in vivo studies [1,2,3,4].

In the last few days of December 2019, a dangerous human-to-human transmitted virus that severely infected the population of Wuhan, China, was reported. The SARS-CoV-2 virus genome and its host cell-interacting key positions have recently been explored. The spike proteins, proteases, angiotensin-converting enzyme-like receptors, and nucleocapsid protein-like targets have been tested to control the disease. COVID-19’s symptoms are characterized by runny nose, coughing, and fever, with an incubation period of 14 days. Recently, blood clotting in the small capsular lungs has been reported in chronic conditions after lung inflammation [5,6]. This detailed exploration made a significant contribution to the development of a diagnostic strategy. Nowadays, molecular techniques such as RT-PCR and immunodiagnostic tools have been developed for the diagnosis of the virus. Plants, naturally derived chemicals, and synthetic chemicals have been tested for control of the disease. Web-based, extensive research on natural proteolytic enzymes involved in blood clotting inhibition has been carried out. The potential inhibitors of the enzyme of SARS-CoV-2 protease, which helps to accelerate COVID-19, were identified by many computational studies. Some of these plants (Ashwagandha), *Ocimum sanctum* (Tulsi), and *Tinospora cordifolia* (Giloy) have different chemical components.

Vaccines against SARS-CoV-2 have been developed by Moderna, BioNTech/Pfizer, Janssen, AstraZeneca, SinoVac, and Gamaleya pharmaceuticals to manage the disease [2,3]. However, the vast worldwide population of concern is still far behind in receiving vaccinations. The immunization program has effectively begun in a number of countries. Additionally, the lack of any available drugs to combat the COVID-19-causing virus compels researchers and medical professionals to look into complementary alternative therapies.

The COVID-19 pandemic is currently a major threat all over the world. To fight the virus, it is essential to achieve and maintain a healthy nutritional state. Several factors, including age, sex, health status, lifestyle, and medications influence an individual’s nutritional status. During the COVID-19 pandemic, individuals’ nutritional status has been used as a source of resistance to instability. The immune system is influenced by optimal nutrition and dietary nutrient intake. Therefore, strengthening the immune system is the only long-term way to live in the current environment. Except for vitamin C, which is one of the best ways to strengthen the immune system, no evidence has established that supplements may cure the immune system. A well-balanced diet can help the body to fight the illness [7,8].

The pineapple (*Ananus comosus*) has been used as folk medicine since antiquity, and the pharmaceutical significance of its active molecule, bromelain, was initially explored since 1975. It has been tested for its anticancer and anti-inflammatory properties, and it inhibits platelet aggregation, fibrinolytic activity, and skin debridement. The anti-inflammatory and platelet aggregation effects are likely linked to its proteolysis activity. Moreover, when there is more plasmin in the blood, it breaks down fibrin into “prohibitory” prostaglandins, which then bind to the PG receptor, which causes adenyl cyclase to work and make cyclic AMP [9]. We give this benefit to patients through active constituents, which are symptoms of COVID-19 disease that need to be taken care of. It will be difficult in a short amount of time to test many bioactive compounds with antimicrobial inhibitory potential against new coronaviruses as well as bacteria. Docking is a quick and economical method for prioritising or choosing possible drug-like compounds for experimental trials. Thus, docking and simulation studies were conducted to find a preventive and potential therapeutic antiviral and antibacterial agent as soon as possible [10].

Gram-negative as well as Gram-positive pathogenic bacteria are an increasing source of serious illnesses and are an urgent concern in medical conditions. These bacterial pathogens are generally non-fermentive, can infect both patients undergoing treatment and others with diverse underlying illnesses or disorders who are not in a therapeutic environment [11]. Thus, for therapeutic purposes, the antibacterial and antiviral potential of bromelain against bacteria and COVID-19 were evaluated.

## 2. Materials and Methods

### 2.1. Drug Compounds Preparation

The 2-dimensional (2D) structures of the drug Artemisinin and natural compound Bromelain (Table 1) were retrieved from PubChem web resource of National Center for Biotechnology Information (NCBI) (https://pubchem.ncbi.nlm.nih.gov/ accessed on 2 March 2022). The 2D chemical structure of the drug was converted to .*pdb* format to run computational based docking analysis and CHARMm forcefield to minimize the files [8,9] using the energy minimization procedure of BIOVIA, Dassault Systèmes, Discovery Studio Visualizer, Version 2020, San Diego, CA, USA [12,13].

### 2.2. Target Molecules Preparation

We have downloaded the crystal 3D structures of SARS-CoV-2 Protease (PDB:6LU7) (Figure 1a) and bacterial receptors, the Crystal structure of *S. aureus* Tyrosyl-tRNA synthetases (TyrRS) (PDB:1JIJ) (Figure 1b), the crystal Structure of *E. coli* 24kDa Domain (PDB:1KZN) (Figure 1c) and *Staph. aureus* DHFR (PDB:3FYV) (Figure 1d) from Protein Data Bank (PDB) (www.rcsb.org accessed on 5 March 2022) [14,15,16,17]. In order to prepare the downloaded 3D crystal structure for docking studies, we have removed the water molecules and HETATM from the published structures and CHARMm force field applied for energy minimization [12,13]. We also analyzed the active site of the pre-bounded ligand molecules in downloaded native structures and obtained the amino acid residue information available in the active site to implement docking analysis by Discovery Studio Visualizer 2020.

### 2.3. In Silico Interaction Analysis

MGL tools 1.5.6, AutoDock 4.2, The Scripps Research Institute, La Jolla, California, USA as used to predict the binding affinity between drug compounds and SARS-CoV-2 Protease and bacterial receptors. Lamarckian genetic algorithm (LGA) was used for interaction studies. Molecular docking methods follow the scoring function by probing the best conformation of receptor-ligand complex on the calculation of binding energy (∆G) by the following equation.
∆G binding = ∆Ggauss + ∆Grepulsion + ∆Ghbond + ∆Ghydrophobic + ∆Gtors

Here, ∆G gauss—attractive term for dispersion of two Gaussian functions; ∆Grepulsion—square of the distance if it is closer than a threshold value; ∆Ghbond—ramp function, also used for interactions with metal ions; ∆Ghydrophobic—ramp function; ∆Gtors—proportional to the number of rotatable bonds [18,19,20].

Further, water molecules were removed from the selected 3D native structures before docking, and hydrogen atoms, Gasteiger charge, Kollman united charges, and solvation parameters were added. The values of the Grid box were set to 60 × 60 × 60° in the X, Y, and Z-axis of a grid point. For the default grid points, spacing was 0.375 Å. *Lamarckian Genetic Algorithm* (LGA) [21,22] was used for drug–protein molecules flexible docking analysis. The default LGA parameters such as population size 150 (gapop_size), energy evaluations 2,500,000 (ga_num_generation), mutation rate 27,000, crossover rate, 0.02 and step size were set to 0.8 and 0.2 Å. The LGA runs were set at 10 runs. After successful execution of docking steps, obtained conformations of receptor–ligand complexes were analyzed for the interactions and binding energy using Discovery Studio molecular visualization software 2020 [14]. Similarly, the interactions of the bromelain peptide with bacterial pathogens *S. aureus* TyrRS (PDB:1JIJ), *E. coli* (PDB:1KZN), and *Staph. aureus* DHFR (PDB:3FYV) targets were tested to see whether it had antibacterial potential, which was observed as binding energy.

### 2.4. Molecular Dynamics Simulation (MDS) Experimentation

The Bromelain–Protease and *S. aureus* TyrRS–bromelain complexes docking results need to be further evaluate through an advanced computational technique. Therefore, an MDS environment was set to execute 100 nanosecond (ns) simulation for both complexes; also, we performed simulation for protease and *S. aureus* TyrRS simulation in water for results comparison using GROningen MAchine for Chemical Simulations (GROMACS) tool 2018 version [23,24] developed by University of Groningen, Netherlands.

*pdb2gmx* module was used to generate required protease (PDB:6LU7) and *S. aureus* TyrRS (PDB: 1JIJ) topology file followed by CHARMM27 all-atom force field selection. In the next step, ligand (bromelain) topology files were generated from Swiss Param server [24]. For the solvation step, a unit cell triclinic box filled with water was created. Addition of Na^+^ and Cl^−^ ions were completed for the stabilization of the system followed by energy minimization. The equilibrium setup of the system (protease–bromelain complex) was required and it was completed, followed by two-step ensembles NVT (constant number of particles, pressure, and temperature) and NPT (constant number of particles, pressure, and **t**emperature). Both ensembles provide control over temperature, pressure coupling, resulting constancy, and stabilization of the system through complete simulation [25]. GORMACS contain several packages, for protease–bromelain complex MDS analysis, we used *gmx rms* for Root Mean Square Deviation (RMSD) [26], *gmx rmsf* for Root Mean Square fluctuation (RMSF), *gmx gyrate* for the calculation of Radius of Gyration (Rg) [22,27], and *gmx hbond* for the calculation of numbers of hydrogen-bond formed during interaction.

Computational prediction of the absorption, distribution, metabolism, excretion and toxicity (ADMET), and pharmacokinetics properties of bromelain compound was performed at pkCSM online server (http://biosig.unimelb.edu.au/pkcsm/ accessed on 22 July 2022) [28].

## 3. Results and Discussion

People who diagnose respiratory tract infections with viruses such as the flu are more likely to develop co-infections, which results in increased disease severity and more deaths. Many different antibiotics, including azithromycin, have been used for the prevention and treatment of bacterial co-infection and subsequent bacterial infections in COVID-19 patients. Antibiotics do not directly affect SARS-CoV-2, but bacterial pneumonia is frequently brought on by viral respiratory infections. Thus, this study was conducted to explore the antibacterial and SARS-CoV-2 antiviral significance of bromelain targeting Gram-positive and Gram-negative bacterial targets and SARS-CoV-2 viral protease [29].

### 3.1. S. aureus TyrRS-Bromelain Docking Analysis

We downloaded the crystal 3D structures of SARS-CoV-2 protease (PDB:6LU7) and bacterial receptors the crystal structure of *S. aureus* TyrRS in complex with SB-239629 (PDB:1JIJ), the crystal Structure of *E. coli* 24 kDa domain in complex with Clorobiocin (PDB:1KZN) and *Staph. aureus* DHFR complexed with NADPH and AR-102 (PDB:3FYV) from Protein Data Bank (PDB) (www.rcsb.org accessed on 5 March 2022). All selected biomolecules already have ligand molecules interacting with active sites. We have analyzed active site amino acid residues information involved in receptor–ligand interaction after visualization in the Discovery Studio Visualization tool. Furthermore, any bounded ligands and water molecules were edited and removed from the published 3D structures before molecular interaction experimentation. After obtaining the docking data, we have analyzed the active site amino acid residues participated in the interaction with the protease–bromelain complex and found that the selected compounds were docked at the same active sites.

The binding affinity was −7.8 kcal/mol (Table 2).Total seven hydrogen bonds were formed between *S. aureus* TyrRS- bromelain interaction. Also, amino acid residues TYR170, ARG88, THR42, HIS47, ASN199, TRP197, GLY198, ASP153, GLN186, GLY193, PRP53, ASP80, ALA39, and ASP40 were involved in hydrophobic interactions formation (Figure 2a,b and Table 2), which were more dominant than the hydrogen bonds. These results have indicated that bromelain interacted more strongly with the Gram-positive bacterial pathogens as compared to the Gram-negative bacterial pathogens.

The antibacterial potential of this protease against *Escherichia coli*, *Vibrio cholera*, *Streptococcus mutans*, *Enterococcus fecalis*, and *Porphyromonas gingivalis* and *Aggregatibacter actinomycetemcomitans* are two common oral pathogens, have also been inhibited [30,31,32].

Tyrosyl-tRNA synthetases (TyrRSs) are ideal cellular sites for therapeutic targets in the management and cure of pathogen attack since they are necessary enzymes as most of the cellular system. Another target for bacteria is dihydrofolate reductase. *S.*
*aureus* was targeted for control via TyrRS and *E. coli* was tested to inhibit via dihydrofolate reductase, a key enzyme which catalyses the synthesis of nucleic acid for microbial cells. The docking results of this study showed that bromelain interacted more strongly with Gram-positive bacterial pathogens than with Gram-negative bacterial pathogens.

Plant protease inhibitors in therapeutics focus on cancer therapy. Previous research has also been reported to inhibit the *E. coli* and *L. monocytogenes* at the concentration of 4 mg/mL which was much more believed to be a significant therapeutic molecule. For significant therapeutic molecules, a molecule potentially inhibits at the lowest conc. Thus, for dose reduction, the researcher reported a cysteine-rich protease from *Bromelia karatas* L. that was isolated, characterized by (LC–MS/MS) and reported to inhibit up to 85% of foodborne bacterial pathogens *S. Typhimurium* and *L. monocytogenes* at a concentration of 10 μg/mL [33,34]. The proteinaceous molecule has differential results against Gram-positive and Gram-negative bacterial pathogens. The binding energy was observed to be more negative with Gram-positive bacterial pathogens as compared to the Gram-negative bacterial pathogens.

Further, the Gram-positive bacterial pathogens were significantly inhibited at low doses as compared to the Gram-negative ones. *S. Typhimurium*, a Gram-negative bacterial pathogen, inhibited at a conc. of 3.0 mg/mL, while Gram-positive *L. monocytogenes* inhibited at the conc of 1.65 mg/mL, supporting the results of conducted studies. Gram-negative bacteria’s external membrane may be resistant to streptococcal pyrogenic exotoxin A (SPEA), whereas Gram-positive bacteria have a peptidoglycan cell wall. The antiviral potential of the molecules was also checked against SARS-CoV-2 protease.

### 3.2. S. aureus TyrRS -Bromelain Simulation Analysis

We have also simulated *S. aureus* target in water and it complexes with bromelain for 100 ns. Further RMSD, RMSF, Rg, and hydrogen bond formation plots were generated and analyzed. The RMSD deviation and fluctuation of the *S. aureus* TyrRS and *S. aureus* TyrRS–bromelain complex showed similar value of 0.2–0.3 nm until 50 ns, but afterward, *S. aureus* TyrRS in water stabled with values of 0.3–0.4 nm while *S. aureus* TyrRS–bromelain complex showed a higher value, of 0.4–0.5 nm, which means that due to presence of bromelain, the *S. aureus* TyrRS backbone were destabilizing (Figure 3a). *S. aureus* TyrRS and *S. aureus* TyrRS–bromelain complex RMSF calculation per residue showed values of 0.1–0.20 nm (Figure 3b) during whole simulation. Major fluctuations were observed at 140–180 and 230–240 amino acid residue regions. Amino acid residues present in these regions were also found in the formation of hydrogen bonds and hydrophobic interactions during docking analysis (Figure 2b). The hydrogen bond plot showed the formation of 1–7 hydrogen bonds during the 1000 ns period (Figure 3c). The observed average value of Rg was between approximately or less 2.0 nm for *S. aureus* TyrRS- in water simulation, while for *S. aureus* TyrRS–bromelain complex showed a value between 2.0–2.05 nm (Figure 3d). Overall, Rg analysis indicated that the *S. aureus* TyrRS–bromelain complex fluctuated until at 40,000 ps, but afterward it remained stable until the end of the simulation.

### 3.3. Protease-Bromelain Docking Analysis

In the search for an effective therapeutic molecule against COVID-19, docking studies with selected molecules with the SARS-CoV-2 virus protease PDB: 6LU7 revealed that bromelain strongly binds to the protease and it can inhibit the virus’ entry into the host cell. The docking results analyzed in this study are represented in Figure 4a,b and Table 3’s results. The observed binding energy for bromelain with protease was −9.37 kcal/mol. The interacting amino acids were THR25, HIS41, SER46, MET49, PHE140, LEU141, ASN142, GLY143, SER144, CYS145, HIS163, HIS164, MET165, GLU166, EU167, PRO168, GLN189. The observed vdW + Hbond + desolv Energy −8.85 kcal/mol, Inhibition Constant was 15.46 uM and total ten hydrogen bonds formed during SARS-CoV-2 Protease-Bromelain interaction. The analyzed docking features are represented in Figure 4, Table 3.

From recently conducted studies, hydroxyl chloroquine and hydroxychloroquine represented a choice for relief but not for a cure, and were also described as having less binding energy (−5.1; −5.7 kcal/mol), but this was more than the binding energy of antiviral drugs Oseltamivir, Ritonavir, and Favipiravir [35,36]. Thus, we selected another antimalarial drug, Artemisinin, to screen its inhibitory potential and found it to have −6.94 kcal/mol binding energy with 8.19 uM and formed three hydrogen bonds using amino acids of SARS-CoV-2 protease HIS163, GLU166 and MET165 and amino acid residues HIS41, PHE140, LEU141, ASN142, GLY143, SER144, CYS145, HIS, HIS163, HIS164, MET165, GLU166, HIS172, GLN189 were involved in hydrophobic interaction (Table 3). It can also be stated that this antimalarial drug could also be potentially inhibited by SARS-CoV-2 protease and could be used in the same way as hydroxychloroquine and remdesivir.

### 3.4. Protease-Bromelain Simulation Analysis

The docking interaction of bromelain was found to be more significant with viral protease (PDB:6LU7). Thus, dynamic simulation studies were conducted only for this target of virus. After the successful run of 100 ns dynamics simulation, analyses were accomplished on the basis of obtained data from RMSD, RMSF, Rg, and the formation of a number of hydrogen bond plots analysis. The deviation and fluctuation of the protease–bromelain complex during the whole simulation period are revealed. The observed results are shown in Figure 5. The average RMSD values observed were between 0.2 and 0.3 nm, while protease simulation in water had RMSD values between 0.15–0.2 nm. Protease molecules remained stable during the whole simulation while, from the starting protease–bromelain complex, they showed little fluctuation until 40 ns. Afterward, they were stable until 100 ns, though a small fluctuation was observed at 25–35 ns (Figure 5a). The RMSF calculation per residue shows values between 0.1–0.20 nm (Figure 5b) for protease and the protease–bromelain complex. Few fluctuations were observed at 10–20, 50–80, 150–200, and 260–280 amino acid residue regions. Amino acid residues present in these regions are also found in the formation of hydrogen bonds in hydrophobic interactions during docking analysis (Figure 4b). The hydrogen bond plot showed the formation of 1–7 hydrogen protease–bromelain bonds during the 1000 ns period (Figure 5c). The radius of gyration analysis is very important for the assessment of the compactness and stability of protein structures during the whole simulation period, due to the presence of ligand molecules. The observed average value of Rg was between 2.2–2.25 nm for the protease in water simulation, while for the protease–bromelain complex it was between 2.1–2.15 nm. (Figure 5d). Overall, Rg analysis indicated that the complex fluctuated at 20,000–250,000 ps.

The adsorption, distribution, metabolism and excretion profiles were estimated. The study drug was not found to absorb through intestine and Skin permeability was −2.735. However, water solubility was observed to be −2.87 and blood–brain barrier crossing capacity and CNS permeability values were −2.889. It was safe for liver enzymes CYP3A4, CYP 1A2, CYP 2C19, CYP 2C9, CYP 2D6, and CYP 3A4. The total clearance value was observed to be 1.686 (Table 4). The obtained data from the pkCSM server (http://biosig.unimelb.edu.au/pkcsm/theory accessed on 23 May 2022) revealed that bromelain has no hepatotoxicity, AMES toxicity, and skin sensitisation properties.

“Let food be thy medicine and medicine be thy food,” Hippocrates advised almost 2500 years ago. Nutritional status is influenced by both nutrient consumption and disease occurrence. A well-balanced diet will help you maintain a robust immune system that will help you resist against microbial attacks including viruses. Much research has suggested that strengthening the immune system is the only long-term way to live in the current environment. It is critical to have enough zinc, iron, and vitamins A, B 12, B6, C, and E to keep your immune system in good shape. Except for Vitamin C, there is currently no evidence that any supplement may ‘boost’ our immune system or treat or prevent viral infections. Vitamin C is one of the most important water-soluble vitamins for maintaining a healthy immune system. Vitamin C has a daily-recommended dietary requirement of 90 milligrams per day for men and 75 milligrams per day for women. In order to tackle COVID-19, it is vital to be aware of the precise forms of food that can strengthen our immune systems [7,8]. Dietary guidelines suggest that fruits be supplemented in diets, such as apples, guavas, bananas, cantaloupe melons, strawberries, grapefruit, papayas, pineapples, lemons, oranges, blackcurrants, etc., during the COVID-19 pandemic. Thus, this evidence suggests that fruits and vegetables have many therapeutic molecules like alkaloids, glycosides, phytosterol and many more that have been used to control many microbial and infectious diseases. Thus, pineapple rich in bromelain has been reported to have many pharmacological activities and it could also be helpful in maintaining the good health of COVID-19 patients [37,38,39,40]. The enzyme has been reported to have potential pharmacological activities including antimicrobial, antioxidant, anti-inflammatory, and anticancer activity. It has good intestinal absorption stability at a wide range of pH (4–9), maintains its maximum concentration after 1 h of its administration and maintains its biological activity with a half-life of 6–9 h, 1 h after administration [40]. Moreover, it is believed that the enzyme complex bromelain also has mucolytic properties and could be beneficial to control respiratory inflammation, including inflammation caused by influenza and asthma allergies [1,7]. It is also thought to have properties that help break up and expel mucus. Many pathological microbial diseases are connected to the involvement of protease in infection of organs such as the mouth, skin, lungs, ears, eyes, nose, and other soft tissues and cavities [10,11]. A virus uses protease to facilitate entry into the host cell and break promising proteins used to pack the new virus particle. Thus, the study reported that bromelain interacted with both target bacteria and viruses significantly, but the observed binding energy of bromelain was more significant as compared to protease, indicating that the significant these interactions with both targets could be helpful to manage bacterial and viral infections [40,41]. Indeed, previously conducted in silico-based studies with the other molecules only reported either antibacterial or antiviral potentialities. In this study, the bromelain molecule has been evaluated for both its antibacterial and antiviral activities. The binding energies of this molecule observed with targets were observed to be more significant as compared to many repurposing drugs and naturally derived molecules. The significant stable interaction with the cellular system has been observed with simulation studies of the molecule, which supports the more significant antimicrobial potentialities of this molecule.

## 4. Conclusions

Pineapple has bromelain, which has many significant clinical benefits in cancer, diabetes, cardiovascular and lung diseases. The analyzed results of bioactive compounds of bromelain from pineapple show that it has the capability to interact with the viral enzyme Main Protease (Mpro) of COVID-19. Bromelain has shown the best binding energy score of the other studied antibiotics against the main protease enzyme of SARS-CoV-2. The docking score of different antibiotics such as tobramycin, ceftriaxone, piperacillin, and penicillin reported previously have inhibitory potential lower than bromelain, which further confirms its significance. Bromelain may have antiviral activities against SARS-CoV-2. It has been previously proven that bromelain is well absorbed in the body after oral administration and has no major side effects, even after prolonged use. Thus, this study suggests that pineapple bromelain could be used as an effective health supplement to control COVID-19 or to synergize the therapeutic effect of other molecules; however, the mechanism of its action has not been well explored until now. Thus, future in vitro and in vivo research can concentrate on investigating the mechanistic therapeutic intervention.

## Figures and Tables

**Figure 1 nutrients-14-03045-f001:**
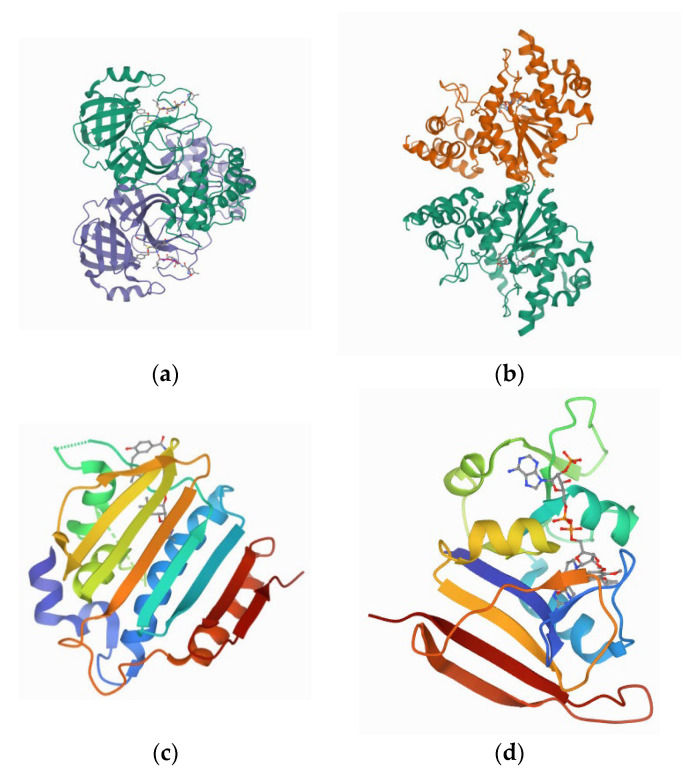
(**a**) the 3D crystal structure of SARS-CoV-2 Protease (PDB:6LU7) (**b**) the crystal structure of *S. aureus* TyrRS (PDB:1JIJ), (**c**) Crystal Structure of *E. coli* 24 kDa Domain (PDB:1KZN) and (**d**) *Staph. aureus* DHFR (PDB:3FYV).

**Figure 2 nutrients-14-03045-f002:**
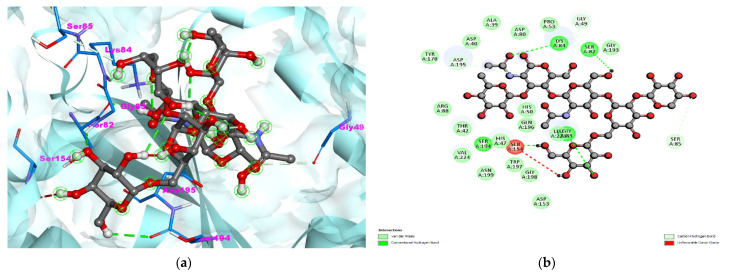
(**a**) 3D visualization of *S. aureus* TyrRS- bromelain complex. The light-turquoise color ribbon pattern shows *S. aureus* TyrRS (PDB:1JIJ) and the bromelain is shown in grey color with ball stick pattern in the center. (**b**) 2D visualization of interaction.

**Figure 3 nutrients-14-03045-f003:**
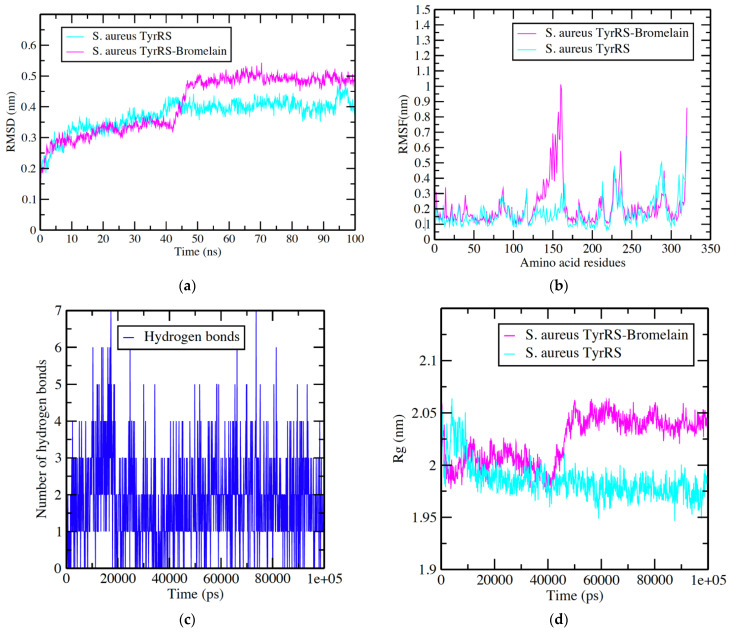
Graphical representation (**a**) RMSD plot of *S. aureus* TyRS in water (Turquois color) and *S. aureus* TyrRS–bromelain complex (Pink color) deviation during 100 ns period. (**b**) RMSF plot with fluctuation per residues. (**c**) Hydrogen bond plot showing formation hydrogen bond (Blue color) during 100,000 ps period. (**d**) Radius of gyration (Rg) plot showing compactness of *S. aureus* TyRS molecule during 100 ns simulation. Where nm = nanometer; ps = picosecond.

**Figure 4 nutrients-14-03045-f004:**
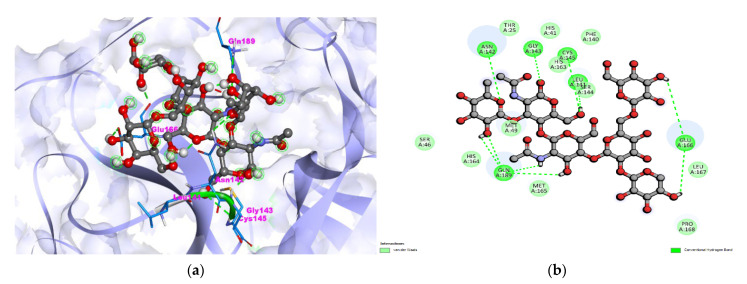
(**a**) Light-blue color ribbon pattern showing SARS-CoV-2 Protease (PDB: 6LU7) interaction with (**b**) bromelain is shown in grey color with ball stick pattern in the center. (**b**) 2D visualization of interaction.

**Figure 5 nutrients-14-03045-f005:**
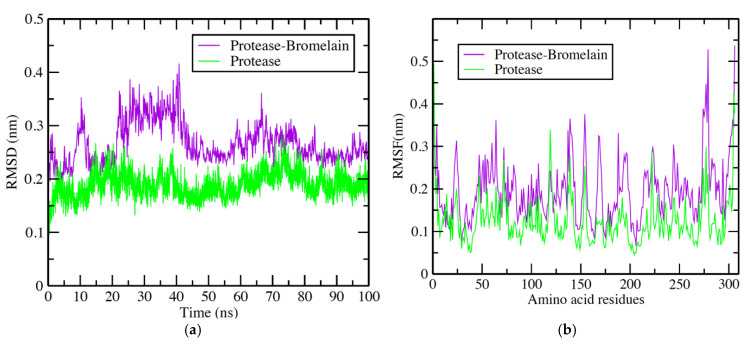
Graphical representation of (**a**) RMSD plot of protease in water (Green color) and protease–bromelain complex (Purple color) deviation during 100 ns period; (**b**) RMSF plot with fluctuation per residue; (**c**) hydrogen bond plot showing formation hydrogen bond (blue color) during 100,000 ps period; (**d**) radius of gyration (Rg) plot showing compactness of protease molecule during 100 ns simulation. Where nm = nanometer; ps = picosecond.

**Table 1 nutrients-14-03045-t001:** Bromelain and selected drug candidate for docking studies against COVID-19 protease.

S.No	Compound Name	Molecular Formula	Molecular Weight	Structure	SMILES ID	PubChem ID
1.	Bromelain	C39H66N2O29	1026.9 g/mol	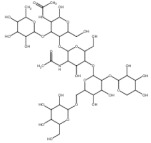	CC1C(C(C(C(O1)OC2C(C(OC(C2OC3C(C(C(C(O3)CO)OC4C(C(C(C(O4)COC5C(C(C(C(O5)CO)O)O)O)O)O)OC6C(C(C(CO6)O)O)O)O)NC(=O)C)CO)O)NC(=O)C)O)O)O	CID: 44263865
2.	Artemisinin	C15H22O_5_	282.33 g/mol	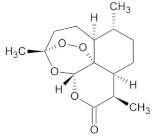	CC1CCC2C(C(=O)OC3C24C1CCC(O3)(OO4)C)C	CID:68827

**Table 2 nutrients-14-03045-t002:** Bromelain docking studies against bacteria receptors. Where in Hydrogen bond details column UNK1 = Bromelain.

PDB IDs	Binding Affinity (Kcal/mol)	Hydrogen Bond Details	Hydrogen Bond Length Angstrom	Hydrophobic Residues/Van der Waals	Other Interaction
1JIJ (Crystal structure of S. aureus TyrRS in complex with SB-239629)	−7.8	A:GLY83:HN–:UNK1:O42	2.90	TYR170,ARG88,THR42,HIS47,ASN199,TRP197,GLY198, ASP153,GLN186,GLY193,PRP53, ASP80,ALA39,ASP40	UNFAVORABLE DONOR = SER154
		A:LYS84:HZ1–:UNK1:O66	2.66		
		:UNK1:H6–A:SER194:O	2.03		
		:UNK1:H91–A:SER82:O	3.70		
		A:SER85:CB–:UNK1:O53	3.47		
		:UNK1:C10–A:ASP195:OD1	3.42		
		:UNK1:C61–A:GLY49:O	3.42		
1KZN (Crystal Structure of *E. coli* 24 kDa Domain in Complex with Clorobiocin)	−6.0	A:GLN135:–:UNK1:O7	2.39	VAL133,MET166,THR165,GLY164, THR160,ILE140,CYS56,GLY75,ARG76,LYS162,ASP74,GLU58,ILE60, ARG206	UNFAVORABLE = LYS57
		:UNK1:H6–A:THR163:OG1	2.89		
		:UNK1:H8–A:HIS55:O	2.30		
		:UNK1:H35–A:GLN72:OE1	2.73		
		:UNK1:C2–A:GLN72:OE1	3.49		
3FYV (*Staph. aureus* DHFR complexed with NADPH and AR-102)	−5.8	X:THR46:HG1–:UNK1:O62	2.00	TRP22,VAL6,ALA7,VAL31,LEU28, LEU5,ILE50,GLN19,LEU62,GLU17, ARG44,LEU97,LYS45,GLY15, PHE98,LEU20,	Pi-sigma = PHE92
		X:GLN95:HN–:UNK1:O56	2.04		
		X:THR96:HG1–:UNK1:O32	2.54		
		X:THR121:HG1–:UNK1:O23	3.01		
		:UNK1:H91–X:ASN18:OD1	2.76		
		:UNK1:H22–X:ILE14:O	2.52		
		:UNK1:H29–X:THR46:O	2.87		
		X:SER49:CB–:UNK1:O12	2.75		
		X:GLY93:CA–:UNK1:O59	3.08		
		X:GLY94:CA–:UNK1:O59	3.11		
		:UNK1:C16–X:ILE14:O	3.65		
		:UNK1:C35–X:THR96:OG1	2.57		
		:UNK1:C61–X:ASN18:O	3.56		
		:UNK1:C61–X:SER49:OG	2.98		

**Table 3 nutrients-14-03045-t003:** Data obtained after performing Molecular interaction between bromelain/selected drug and SARS-CoV-2 protease PDB:6LU7. Where in Hydrogen bond details column UNK1 = compound.

S. No	Compound Name	Final Intermolecular Energy (kcal/mol)	vdW + Hbond + Desolv Energy (kcal/mol)	Electrostatic Energy (kcal/mol)	Inhibition Constant	Hydrogen Bond Details	Hydrogen Bonds Length (Angstrom)	Residues Involved in Hydrophobic Interaction
1.	Bromelain	−9.37	−8.85	−0.51	15.46 uM	A:ASN142:HD22–:UNK1:O7	3.01	Thr25, His41, Ser46, Met49, Phe140, Leu141, Asn142, Gly143, Ser144, Cys145, His163, His164,Met165, Glu166, Leu167, Pro168, Gln189
						A:GLY143:HN–:UNK1:O63	1.84	
						A:CYS145:HN–:UNK1:O62	2.87	
						A:GLN189:HE22–:UNK1:O68	3.03	
						:UNK1:H–A:GLU166:OE1	2.32	
						:UNK1:H–A:GLU166:O	1.91	
						:UNK1:H–A:GLN189:OE1	1.95	
						:UNK1:H–A:GLN189:OE1	2.35	
						:UNK1:H–A:GLN189:OE1	1.82	
						:UNK1:H–A:LEU141:O	1.68	
2.	Artemisinin	−6.94	−6.80	−0.14	8.19 uM	A:HIS163:HE2–:UNK1:O17	1.93	His41,Phe140,Leu141, Asn142,Gly143,Ser144,Cys145,His163,His164,Met165,Glu166,His172,Gln189
						A:GLU166:HN–:UNK1:O9	1.79	
						A:MET165:CA–:UNK1:O18	3.32	

**Table 4 nutrients-14-03045-t004:** ADMET data of bromelain predicted by PKCSM server.

Property	Model Name	Predicted Value	Unit
**Absorption**	Water solubility	−2.87	Numeric (log mol/L)
**Absorption**	Caco2 permeability	−1.262	Numeric (log Papp in 10^−6^ cm/s)
**Absorption**	Intestinal absorption (human)	0	Numeric (% Absorbed)
**Absorption**	Skin Permeability	−2.735	Numeric (log Kp)
**Absorption**	P-glycoprotein substrate	Yes	Categorical (Yes/No)
**Absorption**	P-glycoprotein I inhibitor	No	Categorical (Yes/No)
**Absorption**	P-glycoprotein II inhibitor	No	Categorical (Yes/No)
**Distribution**	VDss (human)	−0.327	Numeric (log L/kg)
**Distribution**	Fraction unbound (human)	0.392	Numeric (Fu)
**Distribution**	BBB permeability	−2.689	Numeric (log BB)
**Distribution**	CNS permeability	−5.75	Numeric (log PS)
**Metabolism**	CYP2D6 substrate	No	Categorical (Yes/No)
**Metabolism**	CYP3A4 substrate	No	Categorical (Yes/No)
**Metabolism**	CYP1A2 inhibitior	No	Categorical (Yes/No)
**Metabolism**	CYP2C19 inhibitior	No	Categorical (Yes/No)
**Metabolism**	CYP2C9 inhibitior	No	Categorical (Yes/No)
**Metabolism**	CYP2D6 inhibitior	No	Categorical (Yes/No)
**Metabolism**	CYP3A4 inhibitior	No	Categorical (Yes/No)
**Excretion**	Total Clearance	1.686	Numeric (log ml/min/kg)
**Excretion**	Renal OCT2 substrate	No	Categorical (Yes/No)
**Toxicity**	AMES toxicity	No	Categorical (Yes/No)
**Toxicity**	Max. tolerated dose (human)	0.377	Numeric (log mg/kg/day)
**Toxicity**	hERG I inhibitor	No	Categorical (Yes/No)
**Toxicity**	hERG II inhibitor	Yes	Categorical (Yes/No)
**Toxicity**	Oral Rat Acute Toxicity (LD50)	2.467	Numeric (mol/kg)
**Toxicity**	Oral Rat Chronic Toxicity (LOAEL)	2.368	Numeric (log mg/kg_bw/day)
**Toxicity**	Hepatotoxicity	No	Categorical (Yes/No)
**Toxicity**	Skin Sensitisation	No	Categorical (Yes/No)
**Toxicity**	*T.Pyriformis* toxicity	0.285	Numeric (log ug/L)
**Toxicity**	Minnow toxicity	28.03	Numeric (log mM)

## Data Availability

Not applicable.

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
