# Peer review of "Exploring the Binding Interaction of Active Compound of Pineapple against Foodborne Bacteria and Novel Coronavirus (SARS-CoV-2) Based on Molecular Docking and Simulation Studies"

_nutrients, 2022, doi:10.3390/nu14153045_

Round 1

Reviewer 1 Report

This is an interesting manuscript that suggests the potential use of natural products derived from the pineapple. However, it is important to mention that the introduction must be improve since nowadays vaccination against SARS-CoV-2 has been accelerated, and the introduction is not up to dated. As well authors should improve their redaction and be more scientific since in the abstract the mention "plants and microbes" like both were the same. I recommend to the authors to review literature regarding on Natural Products and improve their work. 

Regarding on the materials and methods, they seem to be well conducted, however I would like to suggest the use of PBPK models to strength their conclusions.

Author Response

REVIEWER-1

Comments and Suggestions for Authors

This is an interesting manuscript that suggests the potential use of natural products derived from the pineapple. However, it is important to mention that the introduction must be improve since nowadays vaccination against SARS-CoV-2 has been accelerated, and the introduction is not up to dated. As well authors should improve their redaction and be more scientific since in the abstract the mention "plants and microbes" like both were the same. I recommend to the authors to review literature regarding on Natural Products and improve their work.  Regarding on the materials and methods, they seem to be well conducted; however I would like to suggest the use of PBPK models to strength their conclusions.

Reply:# Thanks for comments. The introduction section  and PBPK related information have been improved as per reviewer suggestions.

The Adsorption, distribution, metabolism and excretion profiles were estimated. The study drug was not found to absorb through intestine and Skin permeability was -2.735. However, water solubility was observed to be -2.87 and Blood brain barrier crossing capacity and CNS permeability values were-2.889. it was safe to liver enzymes CYP3A4, CYP 1A2, CYP 2C19, CYP 2C9, CYP 2D6, CYP 3A4. The total clearance value was observed to be .1.686.(Table 4)The Obtained data from pkCSM server (http://biosig.unimelb.edu.au/pkcsm/theory) revealed that Bromelain has no hepatotoxicity, AMES toxicity and skin sensitisation properties.

Reviewer 2 Report

1.       Authors performed docking and dynamic simulation study. Authors need to add computational study content with proper references in the introduction section.

2.       How is the validation of docking conducted? Did the authors compare the active site of their docked structures like the Protease-Bromelain complex, especially with the native structure of protease (PDB:6LU7)? Need to be clarified and discussed properly within the text of the main content.

3.  In table 2 and table 3 no need to mention Hydrogen bond length with such a precise numeric calculation. Just need to correct/mention it as after decimal two digits mathematical counting.

4.Correct the same information in the whole manuscript.

5. A well-balanced diet can help the body fight the illness. Please correct the sentence.

6. A conclusive line needs to be incorporated in the finishing of the introduction.

7. Page 4; Line 119… “Gasteiger charge, Kollman united charges and salvation….”Correct the word salvation; I think the author wants to say salvation.

8. References are not uniform in the body of manuscript and it need to be corrected throughout the manuscript

9. Page 4; line 141….. “topology file followed by CHARMM27 all-atom force field selection”.

10. There are many forcefields available to generate topology files for protein/receptor molecules. The author needs to mention what is the significance of CHARMM27 implementation.

Author Response

Reviewer 2#

  1. Authors performed docking and dynamic simulation study. Authors need to add computational study content with proper references in the introduction section.

Reply: 1#. Thanks for reviewer valuable comments and corrections have been incorporated.

  1. How is the validation of docking conducted? Did the authors compare the active site of their docked structures like the Protease-Bromelain complex, especially with the native structure of protease (PDB:6LU7)? Need to be clarified and discussed properly within the text of the main

Reply 2#. Thanks for your concern. We have download the crystal 3D structures of COVID-19 Protease (PDB ID: 6LU7) and bacterial receptors the Crystal structure of S. aureus TyrRS in complex with SB-239629 (PDB ID: 1JIJ), Crystal Structure of E. coli 24kDa Domain in Complex with Clorobiocin (PDB ID: 1KZN) and Staph. aureus DHFR complexed with NADPH and AR-102 (PDB ID: 3FYV) from Protein Data Bank (PDB) (www.rcsb.org ). All selected biomolecules have already ligand molecules interacting with active sites. We have analyzed active site amino acid residues information involved in receptor-ligand interaction after visualizing in Discovery Studio Visualization tool. Furthermore, any bounded ligands and water molecules were edited and removed from the published 3D structures before molecular interaction experimentation. After obtaining docking data, we have analyzed the active site amino acid residues participated in the interaction with Protease-Bromelain complex and found that selected compounds were docked at the same active sites.

  1. In table 2 and table 3 no need to mention Hydrogen bond length with such a precise numeric calculation. Just need to correct/mention it as after decimal two digits mathematical counting.

Reply#.3  The corrections have been incorporated.

4.Correct the same information in the whole manuscript.

Reply 4#. The corrections have been incorporated.

  1. A well-balanced diet can help the body fight the illness. Please correct the sentence.

Reply 5#. The corrections have been incorporated.

  1. A conclusive line needs to be incorporated in the finishing of the introduction.

Reply 6#: . The lines have been incorporated.

  1. Page 4; Line 119… “Gasteiger charge, Kollman united charges and salvation….”Correct the word salvation; I think the author wants to say salvation.

Reply7#: . Thanks for your suggestion. Corrected accordingly.

  1. References are not uniform in the body of manuscript and it need to be corrected throughout the manuscript

Reply 8#. All references are checked and present as per journal formate.

  1. Page 4; line 141….. “topology file followed by CHARMM27 all-atom force field selection”.

Reply 9#. The correction have been incorporated

  1. There are many forcefields available to generate topology files for protein/receptor molecules. The author needs to mention what is the significance of CHARMM27 implementation.

Reply 10#. Thanks for your concern. CHARMM (Chemistry at HARvard Molecular Mechanics) for molecular simulation is technically most versatile forcefield discovered and used since 30 years. We have chosen and used CHARMM27 version  for docking and simulation because it can implement force-field to each every type of atoms of the different molecules like proteins, peptides, lipids, nucleic acids, carbohydrates and small molecule ligands, as they occur in solution, crystals, and membrane environments. https://www.ncbi.nlm.nih.gov/pmc/articles/PMC2810661/

Reviewer 3 Report

Lack of necessary references in many parts of this article (e.g. line37-41, 42-44, 50-51, 61-63, 70-74). 

Showed not much data for translational research and clinical application.

Bromelain against bacteria and COVID-19 has been reported in many studies.

Krishnan, V. A., & Gokulakrishnan, M. (2015). Extraction, purification of bromelain from pineapple and determination of its effect on bacteria causing periodontitis. International Journal of Pharmaceutical Sciences and Research (IJPSR)6(12), 5284-5294.

Amini, N., Setiasih, S., Handayani, S., Hudiyono, S., & Saepudin, E. (2018, October). Potential antibacterial activity of partial purified bromelain from pineapple core extracts using acetone and ammonium sulphate against dental caries-causing bacteria. In AIP Conference Proceedings (Vol. 2023, No. 1, p. 020071). AIP Publishing LLC.

Ali, A. A., Mohammed, A. M., & Isa, A. G. (2015). Antimicrobial effects of crude bromelain extracted from pineapple fruit (Ananas comosus (Linn.) Merr.).

Kritis, P., Karampela, I., Kokoris, S., & Dalamaga, M. (2020). The combination of bromelain and curcumin as an immune-boosting nutraceutical in the prevention of severe COVID-19. Metabolism open8, 100066.

Sagar, S., Rathinavel, A. K., Lutz, W. E., Struble, L. R., Khurana, S., Schnaubelt, A. T., ... & Radhakrishnan, P. (2020). Bromelain inhibits SARS-CoV-2 infection in VeroE6 cells. Biorxiv.

Showed no novelty and enough data.

Author Response

REVIEWER-3

Comments and Suggestions for Authors

Comments:Lack of necessary references in many parts of this article (e.g. line37-41, 42-44, 50-51, 61-63, 70-74).

Reply:# Thank you for valuable comments. The references has been checked and incorporated as per reviewer suggestions.

The manuscript has been revised as per reviewer suggestion and it has been improved significantly.

Comments: Bromelain against bacteria and COVID-19 has been reported in many studies.

Reply:# As the reviewer suggested some references about the previous studies on bromeliad, and authers agreed with comments. In this conducted study authors used different targets for management of both bacterial and COVID-19 diseases. Thus, this in-silico based study could be a further extension, in significance of previously conducted studies as antibacterial and antiviral potentialities of bromelian. Moreover, the conducted study could be a mechanistic support for further studies to develop this molecule for translational research and clinical applications.

Reviewer 4 Report

1.       The gram-positive bacterial pathogens were significantly inhibited at a low dose as compared to the gram-negative. The justification is need to be incorporated

2.       How does this molecule claim the potential as compared to other molecules? Need to discuss this point with suitable references?

3.       The authors mentioned several scientific names and terminologies for example S. aureus TyrRS. What is TyrRS/DHFR, it is not mentioned in the whole manuscript and it need to explain

4.       It is suggested that follow the journal's standard policy for abbreviation listing as well as within the main text which will help readers to understand the concept of the study.

5.       Bacteria names should be written in italic throughout the manuscript

6.       S. aureus TyrRS, Staph. aureus DHFR, E. coli 24kDa Domain description needs to be incorporated..

7.       Gasteiger charge, Kollman united charges Lamarckian Genetic Algorithm italic nature need to be checked.

8.       References in the methodology section as not mentioned as per journal format.

9. Minor typo and grammatical errors should be corrected. 

Author Response

Reviewer-4

Comment1# The gram-positive bacterial pathogens were significantly inhibited at a low dose as compared to the gram-negative. The justification is need to be incorporated

Reply 1#. The binding energy observed with gram positive bacterial pathogen are more significance negative which support the stronger inhibitory even at low concentration of molecule against gram positive bacterial pathogens as compare to gram negative bacterial pathogens.

Comment2# How does this molecule claim the potential as compared to other molecules? Need to discuss this point with suitable references?

Reply 2#: The previously conducted insilico based studies with the other molecules only reported with either with antibacterial or antiviral potentialities. In this study the bromolein molecule has evaluated for both antibacterial and antiviral activities. The binding energies of this molecule observed with targets were observed to be more significant as compare to many repurpousing drugs and naturally derived molecules. The significant stable interaction with cellular system has observed with simulation studies of molecule which support the more significance antimicrobial potentialities of this molecule.

Comment 3# The authors mentioned several scientific names and terminologies for example S. aureus TyrRS. What is TyrRS/DHFR, it is not mentioned in the whole manuscript and it need to explain

Reply 3#: Tyrosyl-tRNA synthetases (TyrRSs) are ideal cellular sites for therapeutic targets in the management and cure of pathogen attack since they are necessary enzymes as most of the cellular system. Another target for bacteria is dihydrofolate reductase(DHFR).

Comment4# It is suggested that follow the journal's standard policy for abbreviation listing as well as within the main text which will help readers to understand the concept of the study.

Reply 4#: Correction has been incorporated

Comment5# Bacteria names should be written in italic throughout the manuscript

Reply 5#: Correction has been been incorporated

Comment6# S. aureus TyrRS, Staph. aureus DHFR, E. coli 24kDa Domain description needs to be incorporated..

Reply 6#: Tyrosyl-tRNA synthetases (TyrRSs) are ideal cellular sites for therapeutic targets in the management and cure of pathogen attack since they are necessary enzymes as most of the cellular system. Another target for bacteria is dihydrofolate reductase. S aureus was targeted for control via TyrRS and E.coli was tested to inhibit via dihydrofoliate reeducate, a key enzyme which catalysis the synthesis of nucleic acid for microbial cells. The docking results of this study showed that bromelain interacted more strongly with gram-positive bacterial pathogens than with gram-negative bacterial pathogens

Comment7# Gasteiger charge, Kollman united charges Lamarckian Genetic Algorithm italic nature need to be checked.

Reply 7#: Thanks for your suggestion. Not needed italic format. We have corrected accordingly.

Comment 8# References in the methodology section as not mentioned as per journal format.

Reply 8#: References have been corrected

Comment9# Minor typo and grammatical errors should be corrected. 

Reply 9#: The manuscript has been checked for Minor typo and grammatical errors and corrections have been incorporated.

Round 2

Reviewer 1 Report

The current manuscript meet the suggestions that I performed. However, there are some issues that should be addressed prior to publication. For instance, authors use equally the term COVID-19 to make reference to the SARS-COV-2 virus, please modify accordingly what authors want to express since these are not the same: COVID-19 is the disease elicited by SARS COV 2, hence the compounds target the virus.

In the same way, I suggest authors to be more clear and explain how do they select SARS COV 2 Mprotease since at the beginning of the manuscripts they focused in gram negative bacteria. Please improve the narrative.

Finally, correct the typo from Hipócrates quote "the for thy"

Author Response

Response to Reviewer 1

 Comments and Suggestions for Authors

The current manuscript meet the suggestions that I performed. However, there are some issues that should be addressed prior to publication. For instance, authors use equally the term COVID-19 to make reference to the SARS-COV-2 virus, please modify accordingly what authors want to express since these are not the same: COVID-19 is the disease elicited by SARS COV 2, hence the compounds target the virus.

Reply: Thanks for your concern and suggestions. We have edited term SARS COV 2 and COVID-19 at suitable position within the whole manuscript text highlighted with sky color.

In the same way, I suggest authors to be more clear and explain how do they select SARS COV 2 Mprotease since at the beginning of the manuscripts they focused in gram negative bacteria. Please improve the narrative.

Reply: Thanks for your critical review and suggestions we have added a paragraph explaining the suggested point in Page number 6; Line 182----190 and text are highlighted with sky color.

Finally, correct the typo from Hipócrates quote "the for thy"

Reply: Corrected accordingly.

Reviewer 3 Report

Since this research article has been improved, I believe the new version of this article is acceptable.

Author Response

Response to reviewer 3

Comments and Suggestions for Authors

Since this research article has been improved, I believe the new version of this article is acceptable.

Reply: Thanks for your kind observation and decision.
